Sex differences in absolute and relative changes in muscle size following resistance training in healthy adults: a systematic review with Bayesian meta-analysis

Refalo Martin C. martin.refalo@deakin.edu.au 1
Nuckols Greg 2
Galpin Andrew J. 3
Gallagher Iain J. 4
Hamilton D. Lee 1
Fyfe Jackson J. 1
1 Institute for Physical Activity and Nutrition (IPAN), School of Exercise and Nutrition Sciences, Deakin University , Geelong , Australia
2 Stronger by Science LLC , Raleigh , NC , United States of America
3 Parker University, Human Performance Center , Dallas , TX , United States of America
4 Centre for Biomedicine and Global Health, Napier University , Edinburgh , United Kingdom
Centner Christoph
Electronic publication date: 2025 Feb 25
Publication date: 2025
Volume: 13
Electronic Location ID: e19042
Received 2024 Oct 22; Accepted 2025 Feb 3
Copyright: ©2025 Refalo et al.
Copyright year: 2025
Copyright holder: Refalo et al.
License: This is an open access article distributed under the terms of the Creative Commons Attribution License, which permits unrestricted use, distribution, reproduction and adaptation in any medium and for any purpose provided that it is properly attributed. For attribution, the original author(s), title, publication source (PeerJ) and either DOI or URL of the article must be cited.
License URL: https://creativecommons.org/licenses/by/4.0/

Keywords: Sex difference, Muscle hypertrophy, Gender difference, Muscle size, Resistance training

Funding: The authors received no funding for this work.

==============================
Background

Muscle hypertrophy may be influenced by biological differences between males and females. This meta-analysis investigated absolute and relative changes in muscle size following resistance training (RT) between males and females and whether measures of muscle size, body region assessed, muscle fibre type, and RT experience moderate the results.

Methods

Studies were included if male and female participants were healthy (18–45 years old) adults that completed the same RT intervention, and a measure of pre- to post-intervention changes in muscle size was included. Out of 2,720 screened studies, 29 studies were included in the statistical analysis. Bayesian methods were used to estimate a standardised mean difference (SMD), log response ratio (lnRR) with exponentiated percentage change (Exp. % Change of lnRR), and probability of direction (pd) for each outcome.

Results

Absolute increases in muscle size slightly favoured males compared to females (SMD = 0.19 (95% HDI: 0.11 to 0.28); pd = 100%), however, relative increases in muscle size were similar between sexes (Exp. % Change of lnRR = 0.69% (95% HDI: −1.50% to 2.88%)). Outcomes were minimally influenced by the measure of muscle size and not influenced by RT experience of participants. Absolute hypertrophy of upper-body but not lower-body regions was favoured in males. Type I muscle fibre hypertrophy slightly favoured males, but Type II muscle fibre hypertrophy was similar between sexes.

Conclusion

Our findings strengthen the understanding that females have a similar potential to induce muscle hypertrophy as males (particularly when considering relative increases in muscle size from baseline) and findings of our secondary analyses should inform future research that investigates sex differences in highly trained participants and muscle fibre type-specific hypertrophy.

Introduction

Resistance training (RT) promotes increases in muscle fibre and ultimately whole-muscle cross-sectional area, known as skeletal muscle hypertrophy (Haun et al., 2019). The magnitude of muscle hypertrophy with resistance training may vary between individuals (Hubal et al., 2005), and importantly, may be influenced by biological differences between males and females arising after puberty (Handelsman, 2017). For example, postpubescent males have approximately tenfold higher endogenous testosterone levels compared with typical postpubescent females (Vingren et al., 2010). This difference in basal testosterone is believed to be the primary factor explaining greater average muscle mass in adult males compared to females. For example, in untrained and resistance-trained individuals, biceps brachii and quadriceps cross sectional area (CSA) of females is ∼50–60% and ∼70–80% of CSA in males, respectively (Nuzzo, 2023a). The proportion of type II muscles fibres, which undergo greater hypertrophy than type I fibres (Tesch, 1988), is also greater in males than females (Nuzzo, 2023b). This difference in muscle mass and fibre type distribution may contribute to females having ∼50–60% and ∼60–70% of male upper-body and lower-body strength, respectively (at the group level) (Nuzzo, 2023a).

It has been postulated that males experience greater muscle hypertrophy following RT compared to females, potentially due to factors relating to gene expression (Liu et al., 2010) or the higher levels of testosterone in males compared to females, on average (Vingren et al., 2010). A previous meta-analysis compared muscle hypertrophy outcomes between young to middle-aged males and females (Roberts, Nuckols & Krieger, 2020) and found no statistically significant differences in pre- to post-intervention changes in muscle size; however, this meta-analysis did not differentiate absolute (i.e., raw change in muscle size) and relative (i.e., percentage increase in muscle size from baseline) changes in muscle size. Considering the marked differences in baseline muscle size between males and females (Nuzzo, 2023a), investigating both absolute and relative changes is essential for gaining a deeper understanding into the physiological response to RT in males and females. For example, another meta-analysis (Jones et al., 2021) of studies in older adults (>50 years of age) found absolute increases in muscle size following RT favoured males compared to females, with no statistically significant difference in relative muscle hypertrophy. Additionally, previous meta-analyses (Jones et al., 2021; Roberts, Nuckols & Krieger, 2020) did not include analyses of fibre type-specific muscle hypertrophy, despite evidence suggesting potential sex differences in type I and type II muscle fibre cross-sectional area (fCSA) adaptations following RT (Abou Sawan et al., 2021; Moesgaard et al., 2022). Understanding how muscle fCSA responds to RT in males and females is important for RT prescription that aims to promote fibre type-specific muscle hypertrophy. It is also unclear if the RT experience of participants and the assessment of muscle size (e.g., body region assessed, type of measurement) used influence sex differences in muscle hypertrophy.

This Bayesian systematic review with meta-analysis aimed to extend previous meta-analytic findings by investigating (i) differences in absolute and relative changes in muscle size following RT between young to middle-aged males and females, and (ii) whether key variables (i.e., method of muscle size assessment and individual characteristics) moderate the influence of sex on muscle hypertrophy. We employed a Bayesian approach to data analysis to improve the interpretation of the effect estimate, directly model its uncertainty, and enable the results to be presented with posterior probabilities allowing for meaningful and intuitive inferences (Kruschke & Liddell, 2018). This Bayesian framework provides a nuanced understanding of the data by presenting a full probability distribution of the effect estimate, instead of dichotomizing results based on p-values. This approach is particularly advantageous for investigating muscle hypertrophy, where effect estimates can be small, uncertainty is inherent, and the context-specific probability of an effect estimate is often more informative than binary significance testing. Portions of the text within this full article were previously published as part of a preprint (https://doi.org/10.51224/SRXIV.400).

Materials & Methods

A systematic review and meta-analysis were performed in accordance with the Preferred Reporting Items for Systematic Reviews and Meta-Analyses (PRISMA) guidelines (Page et al., 2021) (see Table S1, Supplemental Digital Content 6). The original protocol was registered with Open Science Framework on the 1st of June 2023 (https://osf.io/trz3y/).

Research question(s)

The research question(s) were defined using the participants, interventions, comparisons, outcomes, and study design (PICOS) framework, as follows. The primary research question was: “What is the estimated difference in muscle hypertrophy following RT between young to middle-aged males and females, in both absolute and relative (%) terms?”. To facilitate the interpretation of this research question, we also investigated whether the assessment of muscle size (i.e., body region assessed, type of measurement used, muscle fibre type) or participant RT experience (years) had a moderating effect on the overall outcome of the meta-analysis.

Literature search strategy

The literature search followed the PRISMA guidelines (Page et al., 2021). Original literature searches of the PubMed, SCOPUS and SPORTDiscus databases were started in May 2023 and completed in June 2023. However, an updated systematic search was conducted in August 2024 returning two studies (Grandperrin et al., 2024; Sterczala et al., 2024). The following search terms were used and adapted for each individual database: “resistance training” OR “resistance exercise” OR “strength training” AND “gender” OR “women” OR “woman” OR “female” OR “sex” OR “sex difference” AND “muscle hypertrophy” OR “muscle size” OR “muscle growth” OR “muscle mass” OR “muscle thickness” OR “cross-sectional area”. Search terms were added using the NOT term to reduce the number of irrelevant studies according to exclusion criteria (e.g., older, elderly, sarcopenia, cancer). The reference list of previous meta-analyses (Jones et al., 2021; Roberts, Nuckols & Krieger, 2020) and the retrieved articles were manually searched, and six additional studies (Hakkinen et al., 1998; Hakkinen et al., 2001; Hurlbut et al., 2002; Ivey et al., 2000; Kojic, Mandic & Ilic, 2021; Kosek et al., 2006) that met the inclusion criteria were identified and subject to the screening process, with full-text review confirming eligibility for inclusion (Fig. 1). Only studies conducted in humans were included.

Figure 1 PRISMA flow chart.

Summary of systematic literature search and article selection process.

Study selection

Covidence software (Veritas Health Innovations, Melbourne, Australia) was used to manage and conduct the systematic study selection process, including the removal of duplicates and the exclusion of ineligible studies at each stage of the screening process. An overview of the article identification process is shown in Fig. 1. The article identification process was completed independently (to reduce any bias during this process) by two authors (MR and JF) with any disagreement resolved through discussion with LH, who acted as a referee to make the final decision. Finally, the authors (MR and JF) reviewed the full text to determine eligibility for inclusion based on the inclusion criteria. If any studies were added through reference checking or manual searching, they were subjected to the same screening process as if they were found in the initial database search.

Inclusion criteria

Studies were included if: (1) participants were apparently healthy young to middle-aged (18–45 years old) adults of any RT experience, (2) the experimental comparison involved male and female participants completing the same RT intervention (e.g., set volume, load, frequency, exercises, proximity-to-failure), and (3) one of the following measures of pre- to post-intervention changes in muscle size were included; (a) muscle thickness, (b) whole-limb or muscle CSA or volume, (c) muscle fCSA, or (d) lean body/fat free mass via dual X-ray absorptiometry (DXA) or bioelectrical impedance analysis (BIA). It is important to note that indirect measures of muscle size (BIA and DXA) may not directly equate to changes in skeletal muscle mass and may be affected by fluid alterations (Bone et al., 2017; Rodriguez et al., 2024). Nonetheless, many studies in the relevant literature assess muscle size with BIA and DXA to infer whole-body changes, which we considered relevant to our target population and research question. Only original research articles (English language) in peer reviewed journals were included. Articles that did not meet these criteria were excluded.

Data extraction

Data extraction was carried out by the principal investigators (MR and JF) to capture key information in a table format (Table 1). The following participant characteristics were extracted: (1) RT status (i.e., untrained, or resistance-trained), (2) age, and (3) sex. The following article characteristics were also extracted: (1) first author, (2) sample size, (3) publication date, (4) intervention groups/protocol outlines and duration, and (5) key findings (i.e., percentage increase in muscle size from pre- to post-intervention and an indication of whether any muscle hypertrophy was statistically different between sexes). Raw data from pre- and post-intervention for muscle hypertrophy outcomes were extracted for meta-analysis (if figures were used instead of numerical data, those data were extracted using Web Plot Digitizer (Version 4.6, CA, USA)). Studies were classified as recruiting ‘resistance-trained’ participants if the participants had any level of RT experience immediately prior to study commencement, whereas studies that involved a RT prohibitory period (e.g., “no RT 6-months prior to study commencement”) were classified as recruiting untrained participants. Considering the absence of detail regarding training status in some studies further classification of training status (e.g., “beginner”, “intermediate”, “advanced”, “highly advanced”) with multiple criteria (Santos Junior et al., 2021) is difficult. Several studies prescribed loads based on repetition maximum (RM) rather than as a percentage of 1-RM, with adjustments made throughout the RT intervention. This variation made it challenging to determine an accurate load for data analysis. Similarly, in some studies, the number of sets per muscle group per week was adjusted across the RT intervention, and due to vague reporting, accurately extracting these values was difficult. Furthermore, set termination methods were often ambiguously described, and given the inconsistencies in the definitions of set failure within the RT literature (Refalo et al., 2022), we refrained from classifying studies based on proximity-to-failure. Consequently, and despite the methods stated in our pre-registration (https://osf.io/trz3y/), we decided to omit RT characteristics (i.e., load, set volume, proximity-to-failure) from our sub-group analyses to ensure our results were not confounded by inaccurate data extraction.

Table 1 Summary of data extraction.

Summary of studies included comparing changes in muscle size from pre-to post-intervention between males and females. Data presented as mean ± SD.

Study	Participants	Age (years)	RT protocol	Duration (sessions/week)	Outcome measure (device; muscle)	Key findings	
Abe et al. (2000)	Males (n= 17)
Females (n= 20)
→ Untrained: No RT 1 year prior	37.7 ± 7.2
41 ± 4.1	3 sets x 8–12 reps
→ 60–70% 1-RM
Exercises: Leg extension, leg curl, chest press, horizontal row, biceps curl, triceps extension	12 weeks (3/week)	Lean mass (DXA; total body)

Muscle thickness (ultrasound; biceps, triceps, chest, quadriceps, hamstrings)	↔Total body lean mass between males (+2.6%) and females (+1.7%)

↔ Muscle thickness between males (+10.3%) and females (+10.8%) for all muscle groups measured	
Abou Sawan et al. (2021)	Males (n = 10)
Females (n = 10)
→ Untrained: No RT 3-months prior	23 ± 4
23 ± 5	4 sets x 10–12 reps
→ 75% 1-RM
Exercises: Leg press, leg extension	8 weeks (3/week)	Muscle fCSA (biopsy + histochemistry; VL)
# Type 1 = 84
# Type II = 92	↑Type I VL fCSA observed in males (+21.1%) versus females (+5.6%) but ↔ Type II VL fCSA between males (+18%) and females (+27.5%)	
Abou Sawan et al. (2022)	Males (n = 10)
Females (n = 10)
→ Untrained: No RT 3-months prior	23 ± 4
23 ± 5	4 sets x 10–12 reps
→ 75% 1-RM
Exercises: Leg press, leg extension	8 weeks (3/week)	Muscle thickness (ultrasound; VL)	↔VL thickness between males (+10.7%) and females (+8.2%)	
Alway et al. (1992)	Males (n= 5)
Females (n= 5)
→ Trained: ≥5 years of RT experience	32.8 ± 4.5
34.8 ± 2.7	3–5 sets x 6–14 reps
→ 60–85% 1-RM∧
Exercises: Biceps curl (multiple variations)	24 weeks (2/week)	Muscle CSA [CT; biceps, flexor (brachialis + biceps)]	Biceps and flexor CSA ↑ for both males (+5.6%) and females (+3.1%)*	
Coratella et al. (2018)	Males (n= 13)
Females (n= 13)
→ Untrained: No RT 6-months prior	21.2 ± 2.6
20.8 ± 3	4 sets x 10 reps
→ 120% 1-RM
Exercise: Leg extensions (eccentric only)	8 weeks (2/week)	Muscle thickness (ultrasound; VL)	↔VL muscle thickness between males (+11.1%) and females (+13%)	
Cureton et al. (1988)	Males (n= 7)
Females (n= 9)
→ Untrained: No RT 6-months prior	24.7 ± 2.1
25.5 ± 2.3	1–3 sets x n reps
→ 70–90% 1-RM
Exercises: Multiple exercises targeting all primary muscle groups	16 weeks (3/week)	Muscle CSA (CT; biceps, quadriceps)	↔Biceps and quadriceps CSA between males (+9.5%) and females (+13.1%) for both RT protocols	
Fernandez-Gonzalo et al. (2014)	Males (n= 16)
Females (n= 16)
→ Untrained: No RT 6-months prior	23 ± 4.8
24 ± 4.9	4 sets x 7 reps
→ 83% 1-RM∧
Exercise: Supine squat (flywheel)	6 weeks (2–3/week)	Lean mass (DXA; thigh)	↔Thigh lean mass between males (+4.6%) and females (+5.4%)	
Grandperrin et al. (2024)	Males (n= 12)
Females (n= 12)
→ Trained: Unknown years of RT experience	27.4 ± 4
29 ± 6	4 sets x 10 reps
→ 70% 1-RM
Exercises: Multiple exercises targeting all primary muscle groups	16 weeks
(3/week)	Lean mass (DXA; total body)	↔Total body lean mass between males (+1.9%) and females (+2%)	
Hakkinen et al. (1998)	Males (n = 10)
Females (n= 11)
→ Untrained: No RT experience	42 ± 2
39 ± 3	3–6 sets x 3–15 reps
→ 50–80% 1-RM
Exercises: Leg press, leg extension	24 weeks (2/week)	Muscle CSA (ultrasound; quadriceps)	Quadriceps CSA ↑ for both males (+5.4%) and females (+9.3%)*
	
Hakkinen et al. (2001)	Males (n = 10)
Females (n= 11)
→ Untrained: No RT experience	42 ± 2
39 ± 3	3–6 sets x 3–15 reps
→ 50–80% 1-RM
Exercises: Leg press, leg extension	24 weeks (2/week)	Muscle fCSA (biopsy + histochemistry; VL)
#Type I = 41
#Type II = 37	VL fCSA ↑ for both males (Type I = +18.9%, Type II = +3.3%) and females (Type I = +22.5%, Type II = +39.2%)*	
Hammarstrom et al. (2020)	Males (n= 16)
Females (n= 18)
→ Trained: Unknown years of RT experience	23.6 ± 4.1
22 ± 1.3	Group A: 1 set x 7–10 reps
→ 75–83% 1-RM∧
Group B: 3 sets x 7–10 reps
→ 75–83% 1-RM∧
Exercises: Leg press, leg extension, leg curl	12 weeks (2–3/week)	Muscle CSA (MRI; quadriceps)	Quadriceps CSA ↑ for both males (+4.4%) and females (+4.2%)*	
Hubal et al. (2005)	Males (n= 243)
Females (n= 342)
→ Untrained: No RT 1 year prior	24.8 ± 6.2
23.9 ± 5.5	3 sets x 6–12 reps
→ 70–85% 1-RM∧
Exercises: Biceps curl (multiple variations)	12 weeks (2/week)	Muscle CSA (MRI; biceps)	↑Biceps CSA observed in males (+19.7%) versus females (+17.6%)	
Hurlbut et al. (2002)	Males (n = 10)
Females (n= 9)
→ Untrained: No RT 6-months prior	25 ± 3.2
26 ± 3	1–3 sets x 12–15 reps
→ 60–70% 1-RM∧
Exercises: Multiple exercises targeting all primary muscle groups	24 weeks (3/week)	Lean mass (DXA; total body)	↔Total body lean mass between males (+2.9%) and females (+3.5%)	
Ivey et al. (2000)	Males (n= 11)
Females (n= 11)
→ Untrained: No RT 6-months prior	25 ± 1
26 ± 1	5 sets x 5–20 reps
→≤85% 1-RM∧
Exercise: Leg extension	9 weeks (3/week)	Muscle volume (MRI; quadriceps)	↑Quadriceps muscle volume observed in males (+12.1%) versus females (+6.3%)	
Kojic, Mandic & Ilic (2021)	Males (n= 9)
Females (n= 9)
→ Untrained: No RT 8-months prior	24.7 ± 2.1
23.3 ± 0.5	3–4 sets x n reps
→ 60–70% 1-RM
Exercises: Biceps curl, Back squat	7 weeks (2/week)	Muscle thickness (ultrasound; biceps)

Muscle CSA (ultrasound; RF, VI, VM, VL)	↔Biceps muscle thickness between males (+13.7%) and females (+21.2%)

↔ Quadriceps CSA between males (+3.9%) and females (+5.9%)	
Kosek et al. (2006)	Males (n= 13)
Females (n= 11)
→ Untrained: No RT 5 years prior	26.2 ± 5
27.9 ± 3.6	3 sets x 8–12 reps
→ 80% 1-RM
Exercises: Back squat, leg press, leg extension	16 weeks (3/week)	Lean mass (DXA; total body)

Muscle fCSA (biopsy + histochemistry; VL)
#Type I = 60
#Type II = 63	Lean mass ↑ for both males (+1.7%) and females (+1.7%)*
VL fCSA Both males (Type I = +25.6%, Type II = +31.5%) and females (Type I = +8.8%, Type II = +22.9%) ↑ VL fCSA*	
Lundberg et al. (2019)	Males (n= 8)
Females (n= 8)
→ Untrained: Recreationally active	∼26 ± 4	Group A: 4 sets x 8–12 reps
→ 70–80% 1-RM∧
Group B: 4 sets x 7 reps (flywheel)
Exercise: Leg extension	8 weeks (2–3/week)	Muscle CSA (MRI; quadriceps)
Muscle volume (MRI; Quadriceps)	Quadriceps CSA ↑ for both males (+6.9%) and females (+8.5%) for both RT protocols*
Quadriceps (proximal and distal) muscle volume ↑ for both males (+7.7%) and females (+7.9%) for both RT protocols*	
McMahon et al. (2018)	Males (n= 8)
Females (n= 8)
→ Untrained: No RT 1 year prior	20 ± 2.8
19 ± 8.5	3–4 sets x 8–10 reps
→ 70% 1-RM
Exercises: Back squat, leg press, leg extension, lunge, split squat	8 weeks (3/week)	Muscle pCSA (ultrasound; VL)	↔VL pCSA between males (+22.5%) and females (+30%)	
Moesgaard et al. (2022)	Males (n= 12)
Females (n= 12)
→ Untrained: No RT 1 year prior	28 ± 7
27 ± 7	3 sets x 8–12 reps
→ 70–80% 1-RM∧
Exercises: Leg press, leg extension	8 weeks (3/week)	Muscle fCSA (biopsy + histochemistry; VL)
#Type I = 191
#Type II = 166	↑Type I VL fCSA observed in males (+22.7%) versus females (+6.3%) but ↔ Type II VL fCSA between males (+29%) and females (+25.8%)	
Nunes et al. (2020)	Males (n= 25)
Females (n = 10)
→ Untrained: No RT 6-months prior	∼23.7 ± 5.3	3 sets x 8–12 reps
→ 70–80% 1-RM∧
Exercises: Biceps preacher curl	10 weeks (3/week)	Muscle thickness (ultrasound; biceps)	Biceps thickness ↑ for both males (+10.5%) and females (+8%)*	
O’Hagan et al. (1995)	Males (n= 6)
Females (n= 6)
→ Untrained: No RT experience	21.2 ± 1.2
20 ± 0.8	3–5 sets x 8–12 reps
→ 70–80% 1-RM∧
Exercises: Biceps curl variations	20 weeks (3/week)	Muscle CSA [CT; flexor (brachialis + biceps)]	↔Flexor CSA between males (+13.8%) and females (+26.9%)	
Peterson et al. (2011)	Males (n= 43)
Females (n= 40)
→ Untrained: No RT 1 year prior	∼25.1 ± 5.5	3 sets x 6–12 reps
→ 70–85% 1-RM∧
Exercises: Biceps curl (multiple variations)	12 weeks (2/week)	Muscle volume (MRI; biceps)	↑Biceps muscle volume observed in males (+15.2%) versus females (+12.1%)	
Psilander et al. (2019)	Males (n= 9)
Females (n = 10)
→ Untrained: No RT experience	∼25 ± 1	3 sets x 5–12 reps
→ 70–85% 1-RM
Exercises: Leg press, leg extension	12 weeks (3/week)	Muscle thickness (ultrasound; VL)
Muscle fCSA (biopsy + histochemistry; VL)
#Type I = 198
#Type II = 374	VL thickness ↑ for both males (+9.8%) and females (+9.5%)*
VL fCSA ↑ for both males (+15.1%) and females (+22.6%)*	
Reece et al. (2023)	Males (n= 14)
Females (n= 16)
→ Untrained: No RT 1 year prior	21.5 ± 2.3
22.1 ± 3.6	Group A: 3 sets x 8–12 reps
→ 80% 1-RM
Group B: 3 sets x n reps (BFR)
→ 30% 1-RM
Exercise: Leg extension	6 weeks (3/week)	Muscle CSA (ultrasound; VL)
Muscle fCSA (biopsy + histochemistry; VL)
#Type I = 38
#Type II = 55	VL CSA ↑ for both males (+5.3%) and females (+7.1%) for both RT protocols*
VL fCSA ↑ for both males (Type I = +18.9%, Type II = +17.3%) and females (Type I = +11.3%, Type II = +21.3%) for both RT protocols*	
Ribeiro et al. (2014)	Males (n= 30)
Females (n= 34)
→ Untrained: No RT 6-months prior	22.7 ± 4.4
22.7 ± 4.1	3 sets x 8–12 reps
→ 70–80% 1-RM∧
Exercises: Multiple exercises targeting all primary muscle groups	16 weeks (3/week)	Skeletal muscle mass (BIA; total body)	↔Skeletal muscle mass between males (+4.2%) and females (+3.9%)	
Rissanen et al. (2022)	Males (n= 23)
Females (n= 22)
→ Trained: ≥1 year of RT experience	26.4 ± 3.9
25.5 ± 3.8	Group A: 2–5 sets x 20% VeL
→ 65–75% 1-RM
Group B: 2–5 sets x 40% VeL
→ 65–75% 1-RM
Exercise: Back squat	8 weeks (2/week)	Muscle CSA (ultrasound; VL)	↔VL CSA between males (+17.1%) and females (+21.5%) for both RT protocols	
Schwanbeck et al. (2020)	Males (n= 15)
Females (n= 21)
→ Trained: >2 year of RT experience	∼22.5 ± 3.5	Group A: 3–4 sets x 4–10 reps (free weights)
→ 75–90% 1-RM∧
Group B: 3–4 sets x 4–10 reps (machines)
→ 75–90% 1-RM∧
Exercises: Biceps curl variations, back squat, lunge	8 weeks (1/week)	Muscle thickness (ultrasound; biceps, quadriceps)	↔Biceps and quadriceps muscle thickness between males (+5.4%) and females (+4.5%) for both RT protocols	
Sterczala et al. (2024)	Males (n= 19)
Females (n= 14)
→ Trained: Unknown years of RT experience	28 ± 4
26 ± 5	3–5 sets x 3–10 reps
→ 64–88% 1-RM
Exercises: Multiple exercises targeting all primary muscle groups	12 weeks (3/week)	Lean mass (DXA; total body)
Muscle fCSA (biopsy + histochemistry; VL)
#Type I = N/A
#Type II = N/A	Lean mass ↑ for both males (+3.5%) and females (+3.4%)
↑ VL fCSA in males (Type I = +14.2%, Type II = +7.9%) versus females (Type I = −6%, Type II = −4.2%)	
Walsh et al. (2009)	Males (n= 280)
Females (n= 412)
→ Untrained: No RT 1 year prior	∼24.8 ± 9
∼24 ± 6	3 sets x 6–12 reps
→ 65–90% 1-RM
Exercises: Biceps curl (multiple variations)	12 weeks (2/week)	Muscle CSA (MRI; biceps)	Biceps CSA ↑ for both males (+19.7%) and females (+17.7%)*	
Notes.

BB barbell

BFR blood flow restriction

BIA bioelectrical impedance analysis

CSA cross-sectional area

CT computed tomography

EF elbow flexor

fCSA fibre cross-sectional area

MRI magnetic resonance imaging

pCSA physiological cross-sectional area

RF rectus femoris

Reps repetitions

RM repetition maximum

RT resistance training

sessions/week sessions per muscle group per week

VeL velocity loss

VL vastus lateralis

↑ increased

↓ decreased

↔ no difference between sexes

* results of statistical comparison between sexes not reported.

∧ relative load estimated from repetitions at % of 1-RM chart.

# mean number of muscle fibres analysed for each participant across timepoints.

Methodological quality assessment

Evaluation of methodological study quality (including risk of bias) was conducted (by MR) using the tool for the assessment of study quality and reporting in exercise (TESTEX) scale (Smart et al., 2015). Any ambiguities in methodological quality assessment were resolved by discussion between MR and JF. The TESTEX scale is an exercise science-specific scale used to assess the quality and reporting of exercise training trials. The scale contains 12 criteria that can either be scored a ‘one’ or not scored at all; 1, eligibility; 2, randomisation; 3, allocation concealment; 4, groups similar at baseline; 5, assessor blinding; 6, outcome measures assessed in 85% of patients (three possible points); 7, intention-to-treat; 8, between-group statistical comparisons (two possible points); 9, point-estimates of all measures included; 10, activity monitoring in control groups; 11, relative exercise intensity remained constant; 12, exercise parameters recorded. The best possible total score is 15 points.

Statistical analysis

To provide a more flexible modelling approach and enable results to be interpreted intuitively through reporting of probabilities (Kruschke & Liddell, 2018), we carried out a Bayesian meta-analysis using the “brms” (Bürkner, 2017) package in R (v 4.0.2; R Core Team, 2020). Detailed statistical analysis procedures can be found on the Open Science Framework (https://osf.io/trz3y/). Posterior draws were extracted using “tidybayes” (Kay, 2024) and effect estimates calculated using “emmeans” (Lenth, 2025). The absolute (mean and standard deviation) changes in muscle size from pre- to post-intervention for both male and female participants were extracted from each study. Standardised mean differences were calculated using the pooled standard deviation of baseline values as the denominator (via the “escalc” function in the “metafor” (Viechtbauer, 2010) package) to provide a more balanced estimate of variability between males and females (Morris, 2008). Pooling the standard deviations accounts for variability in both groups and avoids bias from using group-specific standard deviations. This method ensures a more robust comparison, especially when baseline variability differs between groups. Converting absolute values to relative changes for SMD calculation may not be statistically efficient (Vickers, 2001) so we therefore calculated the log response ratio (lnRR) for an interaction effect of group × time factorial design (Lajeunesse, 2015). To enhance practical interpretation, we exponentiated the lnRR values with a correction factor for transformation bias (Spake et al., 2023), thereby converting them to percentage change scores (Exp. % Change of lnRR). Positive values indicate greater muscle size increases in males, and negative values indicate greater increases in females. The Bayesian hierarchical analysis accounted for nested observations and used “shrinkage” to adjust study-level effects towards the overall mean (Kruschke & Liddell, 2018). Shrinkage-adjusted effect estimates are presented (see raw estimates in Fig. S1, Supplemental Digital Content 4). Due to the lack of reported correlations between pre- and post-test measures, we assumed a correlation coefficient of r = 0.87 from a recent meta-analysis (Jones et al., 2021) and conducted sensitivity analyses using r values from 0.7 to 0.99 (see Table S1, Supplemental Digital Content 1). Non-informative priors were used, and inferences were drawn from posterior distributions via Hamiltonian MCMC and highest density intervals (HDI). Interpretations were based on the size of the mean effect estimate (Swinton & Murphy, 2022), HDI limits (Swinton & Murphy, 2022), and the posterior probability (ranging from 50% to 100%) that an effect estimate goes in a particular direction (pd) (Kelter, 2023). Publication bias was visually assessed using funnel plots.

Results

Search results and study characteristics

A total of 30 studies met the inclusion criteria. A PRISMA diagram of the systematic literature search and study selection process is displayed in Fig. 1. Data from one study (Hammarstrom et al., 2020) could not be retrieved; the remaining 29 studies were systematically reviewed and meta-analysed. Visual inspection of funnel plots (see Fig. S1, Supplemental Digital Content 2) identified no publication bias. A total of 1,278 male and 1,537 female data points were included in the meta-analysis, with the mean age of males being 26 ± 4 (range: 20 to 42) and females also 26 ± 4 (range: 19 to 41) years. Six (Alway et al., 1992; Grandperrin et al., 2024; Hammarstrom et al., 2020; Rissanen et al., 2022; Schwanbeck et al., 2020; Sterczala et al., 2024) out of the 29 studies involved participants with some RT experience, with the remainder of the studies involving participants with either (i) no RT experience (n = 4), or (ii) no RT experience 5-years (n = 1), 1-year (n = 7), 8-months (n = 1), 6-months (n = 7), and 3-months (n = 2) prior to study commencement. However, in some cases the exact RT experience (years) of the ‘resistance-trained’ participants was vaguely described and therefore unclear (Table 1). In total, 68 muscle hypertrophy outcomes were extracted, with some studies reporting numerous direct outcomes: (i) muscle CSA using magnetic resonance imaging (MRI) (Hammarstrom et al., 2020; Hubal et al., 2005; Lundberg et al., 2019; Walsh et al., 2009), ultrasound (Hakkinen et al., 1998; Kojic, Mandic & Ilic, 2021; Reece et al., 2023; Rissanen et al., 2022), or computed tomography (CT) (Alway et al., 1992; Cureton et al., 1988; O’Hagan et al., 1995), (ii) muscle fCSA using biopsy samples (Abou Sawan et al., 2021; Hakkinen et al., 2001; Kosek et al., 2006; Moesgaard et al., 2022; Psilander et al., 2019; Reece et al., 2023; Sterczala et al., 2024), (iii) muscle physiological CSA using ultrasound (McMahon et al., 2018), (iv) muscle volume using MRI (Ivey et al., 2000; Lundberg et al., 2019; Peterson et al., 2011), and (v) muscle thickness using ultrasound (Abe et al., 2000; Abou Sawan et al., 2022; Coratella et al., 2018; Kojic, Mandic & Ilic, 2021; Nunes et al., 2020; Psilander et al., 2019; Schwanbeck et al., 2020), and other studies using indirect outcomes: (i) lean mass using DXA (Abe et al., 2000; Fernandez-Gonzalo et al., 2014; Grandperrin et al., 2024; Hurlbut et al., 2002; Kosek et al., 2006; Sterczala et al., 2024), and (ii) estimated skeletal muscle mass using bioelectrical impedance analysis (BIA) (Ribeiro et al., 2014). Most of the muscle hypertrophy outcomes were assessed in the lower body (69% of outcomes (Abe et al., 2000; Abou Sawan et al., 2021; Abou Sawan et al., 2022; Coratella et al., 2018; Cureton et al., 1988; Fernandez-Gonzalo et al., 2014; Hakkinen et al., 1998; Hakkinen et al., 2001; Hammarstrom et al., 2020; Ivey et al., 2000; Kojic, Mandic & Ilic, 2021; Kosek et al., 2006; Lundberg et al., 2019; McMahon et al., 2018; Moesgaard et al., 2022; Psilander et al., 2019; Reece et al., 2023; Rissanen et al., 2022; Schwanbeck et al., 2020; Sterczala et al., 2024); quadriceps and hamstrings) versus the upper-body (22% of outcomes (Abe et al., 2000; Alway et al., 1992; Cureton et al., 1988; Hubal et al., 2005; Kojic, Mandic & Ilic, 2021; Nunes et al., 2020; O’Hagan et al., 1995; Peterson et al., 2011; Schwanbeck et al., 2020; Walsh et al., 2009); biceps, triceps, and chest), with 9% of outcomes (Abe et al., 2000; Grandperrin et al., 2024; Hurlbut et al., 2002; Kosek et al., 2006; Ribeiro et al., 2014; Sterczala et al., 2024) assessing lean mass of the upper- and lower-body combined (i.e., total body lean mass). In some instances, studies were excluded from sub-group analyses because (i) outcome measures were only employed in one study (e.g., pCSA (McMahon et al., 2018) and skeletal muscle mass via BIA (Ribeiro et al., 2014)), and (ii) measures of lean mass were not separated into upper- or lower-body (Grandperrin et al., 2024; Hurlbut et al., 2002; Kosek et al., 2006; Ribeiro et al., 2014; Sterczala et al., 2024). The duration of the RT interventions ranged from six to 24 weeks, with a mean of 11 weeks. For a comprehensive summary of other RT characteristics, see Table 1.

Methodological quality

A detailed overview of the methodological quality of included studies was conducted using the TESTEX scale (see Table S1, Supplemental Digital Content 3). Study quality scores ranged from 9 to 12 (out of a possible 15), with mean and median scores of 10. Although each study had some risk of bias, many studies lost points due to (i) no activity monitoring, (ii) no ‘intention-to-treat’ analysis of participants who had withdrawn, and (iii) no reporting of adverse incidents or compliance rate of participants. Overall, a total of 19 out of 29 (66%) studies scored highly (>10) on the TESTEX scale and visual inspection of methodological quality results revealed no impact of study quality on the effect size estimates generated. Considering that all included studies involved a comparison between males and females, no randomisation procedures were required, allocation concealment was not possible, and muscle size differed at baseline, thus, criterion ‘2’ (i.e., “randomisation specified”), criterion ‘3’ (i.e., “allocation concealment”), and criterion ‘4’ (i.e., “groups similar at baseline”) were given one point for every study. Although randomisation of participants into groups was not necessary in the studies included in this systematic review with meta-analysis, studies that involved different RT groups for each sex, and/or a control group, did employ appropriate randomisation procedures (Abe et al., 2000; Lundberg et al., 2019; McMahon et al., 2018; Nunes et al., 2020; O’Hagan et al., 1995; Psilander et al., 2019; Reece et al., 2023; Rissanen et al., 2022; Schwanbeck et al., 2020).

Figure 2 Meta-analysis of standardised mean differences to assess absolute changes in muscle size from pre- to post-intervention between males and females.

Positive values favour greater increases in muscle size for male participants. Point (mean) estimates and 95% high density credible intervals are shown by the point and interval line below each posterior distribution. Red vertical lines represent the point estimate (solid) and width of the highest density credible interval (dotted) for the pooled effect size. Standardised mean differences shown are adjusted towards the overall mean, known as shrinkage. Note: Walsh et al. (2009), Sterczala et al. (2024), Schwanbeck et al. (2020), Rissanen et al. (2022), Ribeiro et al. (2014), Reece et al. (2023), Psilander et al. (2019), Peterson et al. (2011), O’Hagan et al. (1995), Nunes et al. (2020), Moesgaard et al. (2022), McMahon et al. (2018), Lundberg et al. (2019), Kosek et al. (2006), Kojic, Mandic & Ilic (2021), Ivey et al. (2000), Hurlbut et al. (2002), Hubal et al. (2005), Hammarstrom et al. (2020), Hakkinen et al. (2001), Hakkinen et al. (1998), Grandperrin et al. (2024), Fernandez-Gonzalo et al. (2014), Cureton et al. (1988), Coratella et al. (2018), Alway et al. (1992), Abou Sawan et al. (2021), Abe et al. (2000).

Meta-analysis results

Meta-analysis (including all 68 outcomes) of absolute changes in muscle size from pre- to post-intervention (Fig. 2) estimated a 100% probability of superior absolute muscle hypertrophy in males compared to females (SMD = 0.19 (95% HDI: 0.11 to 0.28)). The HDI covers ESs that suggest a negligible to small effect (favouring males), with low to moderate between-study variance identified (τ = 0.09 (95% HDI: 0.01 to 0.20)). This between-study variance represents 45% of the total variance, with additional contributions from the group (I2group = 27%) and observation (I2observation = 13%) levels. Additionally, meta-analysis (including all 68 outcomes) of lnRR to assess relative changes in muscle size from pre- to post-intervention (Fig. 3) estimated similar muscle hypertrophy in males and females (lnRR = 0.01 (95% HDI: −0.01 to 0.03); pd = 74%). The HDI covers ESs that suggest a negligible effect, with negligible between-study variance identified (τ = 0.01 (95% HDI: 0.00 to 0.03)). Exponentiated percentage changes calculated from lnRR also showed similar muscle hypertrophy between males and females (Exp. % Change of lnRR = 0.69% (95% HDI: −1.50% to 2.88%)). Results of secondary sub-group analyses are displayed in Table 2 (and see Fig. S1 Supplemental Digital Content 5). Raw SMD and Exp. % Change of lnRR of meta-analysed studies are displayed in Supplemental Digital Content (see Fig. S1, Supplemental Digital Content 4).

Figure 3 Meta-analysis of log response ratios (converted to exponentiated percentage changes) to assess relative changes in muscle size from pre- to post-intervention between males and females.

Positive values favour greater increases in muscle size for male participants. Point (mean) estimates and 95% high density credible intervals are shown by the point and interval line below each posterior distribution. Red vertical lines represent the point estimate (solid) and width of the highest density credible interval (dotted) for the pooled effect size. Exponentiated log response ratios are adjusted towards the overall mean, known as shrinkage. Note: Walsh et al. (2009), Sterczala et al. (2024), Schwanbeck et al. (2020), Rissanen et al. (2022), Ribeiro et al. (2014), Reece et al. (2023), Psilander et al. (2019), Peterson et al. (2011), O’Hagan et al. (1995), Nunes et al. (2020), Moesgaard et al. (2022), McMahon et al. (2018), Lundberg et al. (2019), Kosek et al. (2006), Kojic, Mandic & Ilic (2021), Ivey et al. (2000), Hurlbut et al. (2002), Hubal et al. (2005), Hammarstrom et al. (2020), Hakkinen et al. (2001), Hakkinen et al. (1998), Grandperrin et al. (2024), Fernandez-Gonzalo et al. (2014), Cureton et al. (1988), Coratella et al. (2018), Alway et al. (1992), Abou Sawan et al. (2021), Abe et al. (2000).

Table 2 Secondary sub-group analyses of body region, assessment of muscle hypertrophy, and resistance training experience, and muscle fibre type.

Effect estimates displayed as standardised mean difference or exponentiated percentage change of log response ratio. Positive values indicate larger increases in muscle size for male participants.

Categorical variable	Effect estimate	HDI	pd	Obs.	
Absolute change in muscle size (standardised mean difference)		
Body region					
Lower Body	0.17	0.056 to 0.29	100%	47	
Upper Body	0.30	0.14 to 0.44	100%	15	
Assessment of muscle hypertrophy	
Lean Mass	0.02	−0.19 to 0.23	60%	6	
Muscle CSA	0.19	0.04 to 0.34	99%	21	
Muscle fCSA	0.29	0.11 to 0.47	100%	16	
Muscle Thickness	0.19	−0.02 to 0.39	97%	17	
Muscle Volume	0.19	−0.08 to 0.45	92%	6	
Muscle Fibre Type	
Type I	0.39	−0.03 to 0.81	97%	8	
Type II	0.10	−0.33 to 0.52	70%	8	
Resistance training experience	
Resistance-Trained	0.20	0.01 to 0.38	98%	13	
Untrained	0.19	0.09 to 0.29	100%	54	
Relative change in muscle size (exponentiated percentage change)	
Body region					
Lower body	1.04%	−2.03 to 4.2%	75%	47	
Upper body	0.60%	−2.97 to 4.18%	63%	15	
Assessment of muscle hypertrophy	
Lean mass	0.02%	−5.12 to 5.32%	50%	6	
Muscle CSA	0.45%	−3.23 to 4.19%	59%	21	
Muscle fCSA	6.03%	−2.55 to 15.4%	91%	16	
Muscle thickness	0.35%	−3.25 to 3.93%	58%	17	
Muscle volume	2.29%	−8.58 to 14.5%	64%	6	
Muscle fibre type	
Type I	12.7%	−6.84 to 34.9%	90%	8	
Type II	−2.29%	−19.2 to 16.7%	62%	8	
Resistance training experience	
Resistance-trained	0.87%	−3.46 to 5.4%	65%	13	
Untrained	0.66%	−1.79 to 3.07%	71%	54	
Notes.

HDI highest density credible interval

Obs observations

pd probability of direction

Sensitivity analysis

Sensitivity analysis of r values from 0.7 to 0.99 found SMDs between 0.17 and 0.22 (meta-analysis result = 0.19). The primary analysis was conducted with an a priori assumption that the correlation coefficient between pre-test and post-test measures was r = 0.87; this is a reasonable assumption that was obtained from previous literature (Jones et al., 2021), with sensitivity analyses indicating little impact of different correlation coefficient values on the pooled SMD. As such the results of our meta-analysis may be interpreted with increased confidence. Results of sensitivity analysis are displayed in Supplemental Digital Content (see Table S1, Supplemental Digital Content 1).

Discussion

Absolute and relative changes in muscle size

This systematic review with meta-analysis extends previous findings with a total of 29 included studies (versus 10 in a previous meta-analysis (Roberts, Nuckols & Krieger, 2020)), providing an up to date synthesis of the current literature investigating biological sex differences in both absolute and relative muscle hypertrophy following RT. We found absolute increases in muscle size following RT slightly favoured males compared to females (SMD = 0.19), however, the relative increase in muscle size (percentage increase from baseline) following RT was similar between sexes (Exp. % Change of lnRR = 0.69%). Inherent differences in testosterone levels between sexes (Nuzzo, 2023a) are known to be responsible for larger baseline muscle size in males compared to females on average (e.g., out of 68 observations extracted from reviewed studies, only two showed larger baseline muscle size in females). Therefore, differences in absolute muscle hypertrophy that are observed between sexes are likely due to differences in baseline muscle size, whereas relative muscle hypertrophy is based on the proportional increase from baseline size. For example, since females start with less muscle mass on average, the absolute increase will be smaller even if the proportional change (i.e., relative muscle hypertrophy) is similar to that of males. Considering the similar relative increases in muscle size that are observed between sexes, physiological signals (e.g., mechanical tension mediated anabolic signalling, metabolic stress (Wackerhage et al., 2019)) other than sex-specific hormonal balance may play the primary role in promoting muscle hypertrophy following RT (Wackerhage et al., 2019). Supportive of our findings is research highlighting (i) the anabolic properties of estradiol that may contribute to muscle hypertrophy (Chidi-Ogbolu & Baar, 2018; Haizlip, Harrison & Leinwand, 2015; Hansen & Kjaer, 2014), (ii) the positive association between androgen receptor content with muscle hypertrophy (Morton et al., 2018), (iii) similarities in post-exercise protein synthesis and molecular signalling between sexes that triggers muscle hypertrophy (Dreyer et al., 2010; West et al., 2012), and (iv) the acute post-exercise elevation in anabolic hormones does not play a major role in stimulating muscle protein synthesis (Van Every, D’Souza & Phillips, 2024). Taken as a whole, our data suggest RT is likely to induce slightly greater absolute increases in muscle size in males compared to females, but similar relative increases in muscle size from baseline. These findings therefore suggest comparable muscle hypertrophic potential between males and females following RT.

Moderators of absolute and relative changes in muscle size

Sub-group analyses were conducted to assess possible variability in muscle hypertrophy outcomes across measures and body regions. Absolute differences in muscle hypertrophy between sexes were more evident with direct measures (i.e., muscle volume, muscle thickness, and muscle CSA and fCSA) versus indirect measures (i.e., lean mass). However, measures of lean mass should be interpreted with caution as they can be influenced by fluid alterations and may be less accurate versus other direct measures used (Rodriguez et al., 2024). Nonetheless, relative changes in muscle size between sexes were similar across all measures employed. Although sex differences in relative hypertrophy slightly favoured muscle fCSA versus other measures, the very wide HDIs suggests high variability in the response. We categorised body regions measured into either upper- or lower-body and found relative changes in muscle size between sexes were similar independent of the body region assessed. However, absolute changes in muscle size of the upper-body slightly favoured males (SMD = 0.30 versus 0.17), may be attributed to larger baseline differences in muscle size between sexes in the upper- versus lower-body (Nuzzo, 2023a). Overall, these data suggest (i) sex differences in absolute and relative muscle hypertrophy do not appear to depend on the measurement of muscle size, and (ii) males experience slightly greater absolute muscle hypertrophy versus females in body regions where larger baseline differences are evident (i.e., in upper-body versus lower-body muscles).

A total of seven studies (n = 170) using histochemical analysis of skeletal muscle biopsies to determine fCSA (Abou Sawan et al., 2021; Hakkinen et al., 2001; Kosek et al., 2006; Moesgaard et al., 2022; Psilander et al., 2019; Reece et al., 2023; Sterczala et al., 2024) were meta-analysed. Similar to previous findings (Abou Sawan et al., 2021; Moesgaard et al., 2022), we observed a > 90% probability of absolute (SMD = 0.39) and relative (Exp. % Change of lnRR = 12.7%) type I muscle fibre hypertrophy favouring males compared to females, providing further support males have a greater capacity to hypertrophy type I muscle fibres than females. However, the 95% HDIs covered wide effect estimates for both absolute and relative type I muscle fibre hypertrophy, suggesting considerable uncertainty in outcomes. Conversely, we estimated a negligible difference in (i) relative hypertrophy of type II muscle fibres favouring females versus males (Exp. % Change of lnRR = −2.29%; pd = 62%), and (ii) absolute hypertrophy of type II muscle fibres between sexes, despite larger baseline mean muscle fCSA for males (4,616 ± 713 µm2) versus females (3,652 ± 621 µm2) across all studies included in our meta-analysis. These findings support the possibility for sex-specific differences in muscle fibre type hypertrophy. Nonetheless, despite all studies assessing muscle fCSA with histochemical analysis of skeletal muscle biopsies, variability in the number of muscle fibres chosen and subsequently analysed per participant (range = 37 to 374), and how studies reported type II muscle fCSA based on the combination of type IIa and IIx values (which differ in size at baseline and in their physiological response to chronic exercise (Reece et al., 2023)), may have influenced our findings. As such, due to the intricate nature of measuring muscle fCSA in research and the uncertainty and variability in responses observed (see Fig. S1, Supplemental Digital Content 4), our findings should be interpreted with caution and used to inform future research that compares muscle fibre type-specific hypertrophy between males and females.

The RT experience of participants did not seem to influence sex differences in absolute and relative muscle hypertrophy following RT. Previous research has indicated that long-term RT experience alters the physiological response to RT (Damas et al., 2015) and may also cause muscle fibre type transitions that could influence sex-specific muscle hypertrophy (Plotkin et al., 2021). For example, a study in high-level competitive weightlifters (i.e., World/Olympic and National level) found years competing in weightlifting influences the proportion of type II muscle fibres more than biological sex per se, with females having a higher abundance of type II muscle fibres than males (Serrano et al., 2019). Whether a higher proportion of type II muscle fibres in highly trained females would influence sex differences in whole muscle and muscle fibre type-specific hypertrophy remains to be explored. Given only six of the 29 studies included in the meta-analysis involved resistance-trained participants, further research investigating sex differences in muscle hypertrophy within resistance-trained samples is encouraged.

Limitations & future research directions

Although most (66%) of the included studies were of ‘high’ quality, a brief overview of key findings in Table 1 suggests that results are consistent across both low and high quality studies. Our sub-group analysis investigating hypertrophy of type I and type II fibres only involved seven studies with a total of 170 participants, and the wide HDIs highlight the variability in outcomes. Although interpretations about muscle fibre type-specific hypertrophy were based on data from 170 participants, it is possible that a larger pool of evidence may strengthen or weaken the findings. Only six out of 29 studies were conducted in resistance-trained participants, however, in some cases the RT experience (years) of the ‘resistance-trained’ participants was vaguely described and therefore unclear (Table 1). Differences in program design (e.g., volume, intensity, frequency, and exercise selection) and duration across studies also introduces heterogeneity into the analysis, potentially affecting the results. However, results indicate minimal between-study variance, suggesting consistent sex-related effects on hypertrophy across studies. Nonetheless, sources of heterogeneity like RT protocols, exact participant training status, and individual factors (e.g., nutrition, hormonal responses) also produce variance at the group and observation levels. Finally, considering exact correlation coefficient values (between pre-test and post-test measures) could not be retrieved from individual studies, we used an a priori assumption of r = 0.87 to calculate SMDs. However, sensitivity analyses showed minimal impact of varying correlation coefficients on the pooled SMD, supporting the robustness of our meta-analysis results. Future research should investigate sex differences in muscle hypertrophy amongst resistance-trained samples and clearly report RT status of participants (e.g., years of experience). Moreover, the influence of muscle fibre type composition on sex-specific hypertrophy warrants exploration, particularly in highly trained individuals where fibre type distribution may differ from untrained individuals. Future research should also seek to minimise variability in histochemical analyses by standardising methods for measuring muscle fCSA and addressing factors such as the number of fibres analysed per participant. These efforts will enhance the understanding and application of resistance training across populations.

Practical applications

Our findings suggest healthy adult males and females can experience similar muscle hypertrophy following RT, and thus, may experience similar benefits from increases in muscle size. For example, (i) low skeletal muscle mass index is associated with an increased risk of all-cause mortality (Wang et al., 2023), and (ii) some physiological characteristics important for athletic performance (e.g., force production, rate of force development, fatigue resistance) may be influenced by muscle size (Kavvoura et al., 2018; Taber et al., 2019). Considering RT experience may not influence sex differences in muscle hypertrophy, RT programs can follow similar structures for both untrained and resistance-trained males and females, with primary differences in RT prescription based on long-term goals (e.g., aesthetics or performance-based goals) and individual characteristics (e.g., enjoyment, perceptions of discomfort, preferences, stress tolerance, etc). However, potential sex differences in short-term responses to RT, such as neuromuscular fatigue and muscle damage, may be greater in males compared to females (Enns & Tiidus, 2010; Hunter, 2014; Hunter, 2016) and should be considered in RT prescription.

Conclusion

This systematic review with Bayesian meta-analysis investigated differences in muscle hypertrophy following RT between healthy adult males and females. The evidence suggests absolute increases in muscle size following RT slightly favour males (SMD = 0.19), however, relative changes in muscle size are similar between sexes (Exp. % Change of lnRR = 0.69%). These results were not influenced by different measures of muscle size or the RT experience (i.e., untrained or resistance-trained) of participants. Further, differences in absolute muscle hypertrophy favouring males over females were larger in the upper- versus lower-body regions. Although there were possible sex differences in muscle-fibre type specific hypertrophy, with greater type I muscle fibre hypertrophy in males compared to females, our findings should be interpreted with caution due to the intricate nature of measuring muscle fCSA in research and the variability in responses observed. Our primary analyses strengthen the understanding that females have a similar potential to induce muscle hypertrophy as males (particularly when considering relative increases in muscle size from baseline) and findings of our secondary analyses should inform future research that investigates sex differences in highly trained participants and muscle fibre type-specific hypertrophy. Overall, RT may be prescribed similarly for both males and females, with the primary distinctions in RT programming driven by long-term objectives (e.g., aesthetic or performance goals) and individual factors such as enjoyment, personal preferences, and discomfort and stress tolerance.

Supplemental Information

Supplemental Information 1 PRISMA checklist

Supplemental Information 2 How disagreements were resolved

Supplemental Information 3 Supplemental Materials

James Steele is acknowledged for his feedback and advice on the statistical analysis.

Additional Information and Declarations

Competing Interests

Author Contributions

Data Availability

Lee Hamilton, Jackson Fyfe, and Iain Gallagher declare that they have no conflicts of interest or competing interests. Martin Refalo, Greg Nuckols, and Andrew Galpin are all coaches and writers in the fitness industry. Greg Nuckols is the founder of Stronger by Science LLC. No known companies will benefit from the results of the present study.

Martin C. Refalo conceived and designed the experiments, performed the experiments, analyzed the data, prepared figures and/or tables, authored or reviewed drafts of the article, and approved the final draft.

Greg Nuckols conceived and designed the experiments, analyzed the data, authored or reviewed drafts of the article, and approved the final draft.

Andrew J. Galpin conceived and designed the experiments, authored or reviewed drafts of the article, and approved the final draft.

Iain J. Gallagher conceived and designed the experiments, analyzed the data, prepared figures and/or tables, authored or reviewed drafts of the article, and approved the final draft.

D. Lee Hamilton conceived and designed the experiments, authored or reviewed drafts of the article, and approved the final draft.

Jackson J. Fyfe conceived and designed the experiments, performed the experiments, analyzed the data, authored or reviewed drafts of the article, and approved the final draft.

The following information was supplied regarding data availability:

This is a systematic review/meta-analysis.

The code and data are available at Zenodo: REFALO, M. C. (2025). Refalo et al. Sex Differences in Hypertrophy Meta-Analysis [Data set]. In PeerJ. Zenodo. https://doi.org/10.5281/zenodo.14611284.

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
