# Peer review of "Sex differences in absolute and relative changes in muscle size following resistance training in healthy adults: a systematic review with Bayesian meta-analysis"

_PeerJ, doi:10.7717/peerj.19042_

## Round 0.1 · original submission · Major Revisions

· Academic Editor

Major Revisions

We thank the authors for submitting their manuscript which is highly relevant within the literature of resistance training and sex differences. After careful review of expert reviewers, both major and minor comments have been raised which need to be addressed by the authors before potential publication.

·

Basic reporting

SEX DIFFERENCES IN ABSOLUTE AND RELATIVE CHANGES IN MUSCLE SIZE FOLLOWING RESISTANCE TRAINING IN HEALTHY ADULTS A SYSTEMATIC REVIEW WITH BAYESIAN META-ANALYSIS



General Commentary
This article presents a very interesting and pertinent question of research of the investigated absolute and relative changes in muscle size following resistance training between males and females and whether measures of muscle size, body region assessed, muscle fibre type, and resistance training between males and females and whether measures of muscle size, body experience moderate the results.

Congratulations on the excellent manuscript.

MINOR CONSIDERATION
ABSTRACT
Conclusion
My only suggestion is to add the approximate percentage values of the differences found in the conclusion of the abstract.

CONCLUSION
I suggest the authors in the final conclusion, add the percentage values of each variable found by the study.

Experimental design

In Q1

Validity of the findings

In Q1

Additional comments

In Q1

Reviewer 2 ·

Basic reporting

The manuscript is well-written in professional and unambiguous English. However, minor grammatical improvements are recommended. For example:

Replace "males versus females" with "males compared to females" for better readability in multiple sections (e.g., Abstract, Discussion).
Simplify technical jargon in the "Practical Applications" section to make it more accessible to a broader audience.
Background and Context:
The introduction effectively provides a solid background on sex differences in muscle hypertrophy, citing relevant literature. However:

The authors could expand on the rationale for employing Bayesian methods in the context of muscle hypertrophy studies. This would clarify how this approach offers advantages over traditional frequentist methods.
Include more recent studies (if available) on hormonal influences on muscle hypertrophy to ensure a comprehensive literature review.

Experimental design

The research questions are well-defined, addressing a critical gap in understanding sex-specific muscle hypertrophy. However, the manuscript would benefit from emphasizing how this study builds on and diverges from prior meta-analyses

Validity of the findings

Data Robustness:

The statistical analyses are robust, with appropriate sensitivity analyses conducted. However, provide a brief explanation of why the correlation coefficient (r = 0.87) was chosen for SMD calculation.
The authors should elaborate on the potential confounding factors that could have influenced the observed sex differences, particularly in muscle fiber type-specific hypertrophy.
Conclusions:

The conclusions align with the results but could benefit from a more explicit statement on practical applications, especially for training regimens targeting specific demographics (e.g., older adults, elite athletes).
Novelty and Contribution:

The study's novelty is clear in its exploration of both absolute and relative changes in muscle size. Highlighting specific implications for future research in sports science or rehabilitation would enhance its value.

Additional comments

Introduction
Insufficient emphasis on the gap addressed by the study. For example, Lines 61-64 mention absolute vs. relative hypertrophy but do not illustrate their practical implications.
Lines 68-75: The exclusion of muscle fiber-specific data in past meta-analyses is noted but lacks depth on the significance of this gap.
Materials & Methods
Line 91-97: The inclusion of indirect methods (e.g., DXA, BIA) is insufficiently justified, considering their limitations (e.g., sensitivity to fluid shifts).
Line 177-205: The use of r = 0.87 as the assumed correlation coefficient lacks strong justification, and sensitivity analysis results are not presented clearly.
Line 147-150: The classification of training experience is vague, failing to account for varying levels of resistance training exposure.
Line 163-165: The heterogeneity of measurement methods (MRI, DXA, ultrasound, etc.) is not addressed in detail, raising concerns about comparability across studies.
Results
Line 215-221: Study characteristics are listed but not summarized in a way that facilitates understanding.
Line 269-279: Figures (e.g., Figure 3) are clear but lack annotations to help readers interpret findings (e.g., negligible effect sizes).
Line 306-313: Sensitivity analysis is briefly mentioned without sufficient detail on how varying r-values impacted results.
Discussion
Line 316-319: The explanation of absolute and relative hypertrophy differences is overly general and lacks detailed examples.
Line 321-323: The influence of baseline differences in muscle size on absolute but not relative hypertrophy is underexplored.
Line 318: "absolute and relative hypertrophy is observed" → Incorrect subject-verb agreement. Should be: "absolute and relative hypertrophy are observed."
Line 323: "This suggests similar growth potentials" → Vague antecedent for "this." Rewrite for clarity: "These findings suggest that males and females have similar growth potentials despite differences in absolute changes."
Line 349-352: The explanation for greater upper-body hypertrophy in males attributes differences solely to baseline size, neglecting physiological mechanisms.
Line 356-359: The potential role of training histories or habitual activities in influencing sex differences is omitted.
Line 352: "favors males due to baseline size differences" → Slightly informal. Consider: "is attributed to baseline size differences favoring males."
Line 359: "This might also explain why..." → "This" is ambiguous. Rewrite: "The observed differences might also be explained by..."
Line 373-375: Variability in type I and type II fiber hypertrophy is acknowledged but not sufficiently explained.
Line 378-381: Methodological inconsistencies in muscle biopsy techniques are not discussed.
Lines 389-400:
Line 391-393: Limited data on highly trained individuals is noted, but specific strategies for addressing this gap are not provided.
Line 397-400: The potential impact of long-term training on hypertrophy adaptations is not explored.
Line 393: "there is little data available" → "Data" is plural. Should be: "there are few data available."
Line 405-407: The variability introduced by different measurement methods (e.g., DXA, MRI, BIA) is not discussed in sufficient detail.
Line 410-413: Potential biases in indirect methods like BIA are not adequately analyzed.
Line 405: "methods like MRI and DXA are accurate" → Sentence structure suggests equality. Rewrite: "Methods like MRI are considered more accurate than DXA or BIA for assessing hypertrophy."
Line 421-423: Recommendations are generic, offering little actionable guidance for different populations.
Line 428-430: Guidance for women on upper-body training lacks specificity.
Line 423: "Training programs can be similar for males and females" → Ambiguous phrasing. Rewrite: "Resistance training programs can follow similar structures for males and females, focusing on relative hypertrophic potential."
Line 430: "females might need extra focus on upper-body training" → Awkward phrasing. Rewrite: "Females may benefit from additional emphasis on upper-body training to address baseline disparities."
Line 412-415: Limitations are broadly stated without directly connecting them to methodological challenges (e.g., variability in exercise protocols).
Line 430-432: Future research suggestions lack innovation and fail to address key gaps, such as hormonal moderators of hypertrophy.

·

Basic reporting

.

Experimental design

.

Validity of the findings

.

Additional comments

Assessment of Publication Bias: Although this study utilized funnel plots to assess publication bias, this method can be relatively subjective. It is recommended to incorporate more objective statistical tests, such as Egger's test and Begg's test, to further evaluate publication bias.

Evaluation of Study Heterogeneity: While this study employed Bayesian meta-analysis to handle heterogeneity across the included studies, assessing and discussing the potential impact of heterogeneity on the analysis results remains necessary and valuable. Consider using the I² statistic or the Q-test to quantify heterogeneity between studies, and provide an interpretation and discussion of the potential sources of heterogeneity.

Time Span of Included Studies: The included studies span 36 years, during which significant variations in technical standards and guidelines for resistance training may exist. Although the authors have sufficiently categorized different training methods, subgroup analysis was not conducted due to concerns about data extraction and classification bias. However, differences in training duration and frequency across studies should be acknowledged. Despite the standardization of outcome measures, the potential bias introduced by the design of intervention protocols should still be considered.

Consideration of Racial Differences: Differences between racial groups should be taken into account, and the inclusion of subgroup analyses based on race may enhance the clinical relevance and applicability of the findings.

Age Range of Participants: The study explicitly defines the participants as healthy adults aged 18-50 years in the abstract and inclusion criteria. However, considering that the study's primary focus is on sex differences in the effects of resistance training on muscle, it is important to note that estrogen plays a critical role in maintaining muscle mass and strength in females. It is generally recognized that from the age of 45, women gradually enter menopause, leading to significant fluctuations in estrogen levels that could influence the effects of resistance training on muscle outcomes. Nevertheless, since the age range of the included female participants is below 45 years, the impact of menopausal estrogen changes on the study results is minimal. To avoid unnecessary concerns, it may be advisable to revise the inclusion criteria to an age range of 18-45 years. As more studies accumulate, future meta-analyses could consider this factor for a more comprehensive analysis.

Reviewer 4 ·

Basic reporting

The introduction was succinct and provided an appropriate amount of background to justify the current meta-analysis. The only suggestion that I have can be disregarded if you do not find it necessary: If there is data available that supports the use of relative over absolute or vice versa, it could be helpful to mention. My hesitancy for this derives from the fact that increases in muscle mass in the context of this meta are obviously beneficial, but it could be helpful to justify why either relative or absolute increases in muscle mass are important measures. I have not personally come across any data for this, but if there were predefined targets for increases in absolute and relative muscle mass that have been shown to be physiologically beneficial, this could be helpful.

The figures should be re-exported with higher dpi to increase clarity.

Line 327—"since females state with less" should be "start"
To follow up on this point. I would suggest mentioning that, for the same reason, females are biased towards greater relative increases in muscle mass due to the lower starting number. The bias towards males for absolute gains and females for relative gains is the main reason I provided my first comment in this section.

Experimental design

Did the included studies attempt to limit bias during data collection through appropriate fasting windows and/or limiting exercise during the day(s) prior to data collection when appropriate?

I appreciate you all not simply saying that you omitted the program characteristics but providing details as to why this was the case so that future authors can be better aware of the issues that arise when small details are not recorded when conducting a training study. The same can be said regarding training status.

The methods were well described and are very transparent, I have no further suggestions in this area.

Validity of the findings

The authors did a good job in the results and discussion section, linking their data to the original research question. I have no further suggestions in this area.

---

## Round 0.2 · accepted · Accept

· Academic Editor

Accept

We congratulate the authors on a well improved manuscript

Reviewer 2 ·

Basic reporting

The revisions in this manuscript are acceptable.

Experimental design

The revisions in this manuscript are acceptable.

Validity of the findings

The revisions in this manuscript are acceptable.

Additional comments

I believe the manuscript meets the requirements for publication

·

Basic reporting

I do not have any further questions and I support the publication of this paper

Experimental design

I do not have any further questions and I support the publication of this paper

Validity of the findings

I do not have any further questions and I support the publication of this paper

Additional comments

I do not have any further questions and I support the publication of this paper

Reviewer 4 ·

Basic reporting

I have no further comments or concerns.

Experimental design

I have no further comments or concerns.

Validity of the findings

I have no further comments or concerns.

Additional comments

I have no further comments or concerns.